# On Coresets for Logistic Regression

**Alexander Munteanu**
Department of Computer Science
TU Dortmund University
44227 Dortmund, Germany
alexander.munteanu@tu-dortmund.de

**Chris Schwiegelshohn**
Department of Computer Science
Sapienza University of Rome
00185 Rome, Italy
schwiegelshohn@diag.uniroma1.it

**Christian Sohler**
Department of Computer Science
TU Dortmund University
44227 Dortmund, Germany
christian.sohler@tu-dortmund.de

**David P. Woodruff**
Department of Computer Science
Carnegie Mellon University
Pittsburgh, PA 15213, USA
dwoodruf@cs.cmu.edu

## Abstract

Coresets are one of the central methods to facilitate the analysis of large data. We continue a recent line of research applying the theory of coresets to logistic regression. First, we show the negative result that no strongly sublinear sized coresets exist for logistic regression. To deal with intractable worst-case instances we introduce a complexity measure $\mu(X)$, which quantifies the hardness of compressing a data set for logistic regression. $\mu(X)$ has an intuitive statistical interpretation that may be of independent interest. For data sets with bounded $\mu(X)$-complexity, we show that a novel sensitivity sampling scheme produces the first provably sublinear $(1 \pm \varepsilon)$-coreset. We illustrate the performance of our method by comparing to uniform sampling as well as to state of the art methods in the area. The experiments are conducted on real world benchmark data for logistic regression.

## 1 Introduction

Scalability is one of the central challenges of modern data analysis and machine learning. Algorithms with polynomial running time might be regarded as efficient in a conventional sense, but nevertheless become intractable when facing massive data sets. As a result, performing data reduction techniques in a preprocessing step to speed up a subsequent optimization problem has received considerable attention. A natural approach is to sub-sample the data according to a certain probability distribution. This approach has been successfully applied to a variety of problems including clustering [31, 22, 6, 4], mixture models [21, 33], low rank approximation [16], spectral approximation [2, 32], and Nyström methods [2, 37].

The unifying feature of these works is that the probability distribution is based on the sensitivity score of each point. Informally, the sensitivity of a point corresponds to the importance of the point with respect to the objective function we wish to minimize. If the total sensitivity, i.e., the sum of all sensitivity scores $\mathfrak{S}$, is bounded by a reasonably small value $S$, there exists a collection of input points known as a coreset with very strong aggregation properties. Given any candidate solution (e.g., a set of $k$ centers for $k$-means, or a hyperplane for linear regression), the objective function computed on the coreset evaluates to the objective function of the original data up to a small multiplicative error. See Sections 2 and 4 for formal definitions of sensitivity and coresets.

**Our Contribution** We investigate coresets for logistic regression within the sensitivity framework. Logistic regression is an instance of a generalized linear model where we are given data $Z \in \mathbb{R}^{n \times d}$,

and labels $Y \in \{-1, 1\}^n$. The optimization task consists of minimizing the negative log-likelihood $\sum_{i=1}^{n} \ln(1 + \exp(-Y_i Z_i \beta))$ with respect to the parameter $\beta \in \mathbb{R}^d$ [34].

• Our first contribution is an impossibility result: logistic regression has no sublinear streaming algorithm. Due to a standard reduction between coresets and streaming algorithms, this also implies that logistic regression admits no coresets or bounded sensitivity scores in general.

• Our second contribution is an investigation of available sensitivity sampling distributions for logistic regression. For points with large contribution, where $-Y_i Z_i \beta \gg 0$, the objective function increases by a term almost linear in $-Y_i Z_i \beta$. This questions the use of sensitivity scores designed for problems with squared cost functions such as $\ell_2$-regression, $k$-means, and $\ell_2$-based low-rank approximation. Instead, we propose sampling from a mixture distribution with one component proportional to the *square root* of the $\ell_2^2$ leverage scores. Though seemingly similar to the sampling distributions of e.g. [6, 4] at first glance, it is important to note that sampling according to $\ell_2^2$ scores is different from sampling according to their square roots. The former is good for $\ell_2$-related loss functions, while the latter preserves $\ell_1$-related functions such as the linear part of the original logistic regression loss function studied here. The other mixture component is uniform sampling to deal with the remaining domain, where the cost function consists of an exponential decay towards zero. Our experiments show that this distribution outperforms uniform and $k$-means based sensitivity sampling by a wide margin on real data sets. The algorithm is space efficient, and can be implemented in a variety of models used to handle large data sets such as 2-pass streaming, and massively parallel frameworks such as Hadoop and MapReduce, and can be implemented in input sparsity time, $\tilde{O}(\texttt{nnz}(Z))$, the number of non-zero entries of the data [12].

• Our third contribution is an analysis of our sampling distribution for a parametrized class of instances we call $\mu$-complex, placing our work in the framework of *beyond worst-case analysis* [5, 39]. The parameter $\mu$ roughly corresponds to the ratio between the log of correctly estimated odds and the log of incorrectly estimated odds. The condition of small $\mu$ is justified by the fact that for instances with large $\mu$, logistic regression exhibits methodological problems like imbalance and separability, cf. [35, 26]. We show that the total sensitivity of logistic regression can be bounded in terms of $\mu$, and that our sampling scheme produces the first coreset of provably sublinear size, provided that $\mu$ is small.

**Related Work** There is more than a decade of extensive work on sampling based methods relying on the sensitivity framework for $\ell_2$-regression [19, 20, 32, 15] and $\ell_1$-regression [10, 40, 11]. These were generalized to $\ell_p$-regression for all $p \in [1, \infty)$ [17, 44]. More recent works study sampling methods for $M$-estimators [14, 13] and extensions to generalized linear models [27, 36]. The contemporary theory behind coresets has been applied to logistic regression, first by [38] using first order gradient methods, and subsequently via sensitivity sampling by [27]. In the latter work, the authors recovered the result that bounded sensitivity scores for logistic regression imply coresets. Explicit sublinear bounds on the sensitivity scores, as well as an algorithm for computing them, were left as an open question. Instead, they proposed using sensitivity scores derived from any $k$-means clustering for logistic regression. While high sensitivity scores of an input point for $k$-means provably do not imply a high sensitivity score of the same point for logistic regression, the authors observed that they can outperform uniform random sampling on a number of instances with a clustering structure. Recently and independently of our work, [41] gave a coreset construction for logistic regression in a more general framework. Our construction is without regularization and therefore can be also applied for any regularized version of logistic regression, but we have constraints regarding the $\mu$-complexity of the input. Their result is for $\ell_2^2$-regularization, which significantly changes the objective and does not carry over to the unconstrained version. They do not constrain the input but the domain of optimization is bounded. This indicates that both results differ in many important points and are of independent interest.

All proofs and additional plots from the experiments are in the appendices A and B, respectively.

## 2   Preliminaries and Problem Setting

In logistic regression we are given a data matrix $Z \in \mathbb{R}^{n \times d}$, and labels $Y \in \{-1, 1\}^n$. Logistic regression has a negative log-likelihood [34]

$$\mathcal{L}(\beta | Z, Y) = \sum_{i=1}^{n} \ln(1 + \exp(-Y_i Z_i \beta))$$

which from a learning and optimization perspective, is the objective function that we would like to minimize over $\beta \in \mathbb{R}^d$. For brevity we fold for all $i \in [n]$ the labels $Y_i$ as well as the factor $-1$ in the exponent into $X \in \mathbb{R}^{n \times d}$ comprising row vectors $x_i = -Y_i Z_i$. Let $g(z) = \ln(1 + \exp(z))$. For technical reasons we deal with a weighted version for weights $w \in \mathbb{R}^n_{>0}$, where each weight satisfies $w_i > 0$. Any positive scaling of the all ones vector $\mathbf{1} = \{1\}^n$ corresponds to the unweighted case. We denote by $D_w$ a diagonal matrix carrying the entries of $w$, i.e., $(D_w)_{ii} = w_i$, so that multiplying $D_w$ to a vector or matrix has the effect of scaling row $i$ by a factor of $w_i$. The objective function becomes

$$f_w(X\beta) = \sum\nolimits_{i=1}^{n} w_i g(x_i \beta) = \sum\nolimits_{i=1}^{n} w_i \ln(1 + \exp(x_i \beta)).$$

In this paper we assume we have a very large number of observations in a moderate number of dimensions, that is, $n \gg d$. In order to speed up the computation and to lower memory and storage requirements we would like to significantly reduce the number of observations without losing much information in the original data. A suitable data compression reduces the size to a sublinear number of $o(n)$ data points while the dependence on $d$ and the approximation parameters may be polynomials of low degree. To achieve this, we design a so-called coreset construction for the objective function. A coreset is a possibly (re)weighted and significantly smaller subset of the data that approximates the objective value for any possible query points. More formally, we define coresets for the weighted logistic regression function.

**Definition 1** (($1 \pm \varepsilon$)-coreset for logistic regression). *Let $X \in \mathbb{R}^{n \times d}$ be a set of points weighted by $w \in \mathbb{R}^n_{>0}$. Then a set $C \in \mathbb{R}^{k \times d}$, (re)weighted by $u \in \mathbb{R}^k_{>0}$, is a $(1 \pm \varepsilon)$-coreset of $X$ for $f_w$, if $k \ll n$ and*

$$\forall \beta \in \mathbb{R}^d : |f_w(X\beta) - f_u(C\beta)| \leq \varepsilon \cdot f_w(X\beta).$$

$\mu$**-Complex Data Sets** We will see in Section 3 that in general, there is no sublinear one-pass streaming algorithm approximating the objective function up to any finite constant factor. More specifically there exists no sublinear summary or coreset construction that works for all data sets. For the sake of developing coreset constructions that work *reasonably well*, as well as conducting a formal analysis beyond worst-case instances, we introduce a measure $\mu$ that quantifies the *complexity* of compressing a given data set.

**Definition 2.** *Given a data set $X \in \mathbb{R}^{n \times d}$ weighted by $w \in \mathbb{R}^n_{>0}$ and a vector $\beta \in \mathbb{R}^d$ let $(D_w X\beta)^-$ denote the vector comprising only the negative entries of $D_w X\beta$. Similarly let $(D_w X\beta)^+$ denote the vector of positive entries. We define for $X$ weighted by $w$*

$$\mu_w(X) = \sup_{\beta \in \mathbb{R}^d \setminus \{0\}} \frac{\|(D_w X\beta)^+\|_1}{\|(D_w X\beta)^-\|_1}.$$

*$X$ weighted by $w$ is called $\mu$-complex if $\mu_w(X) \leq \mu$.*

The size of our $(1 \pm \varepsilon)$-coreset constructions for logistic regression for a given $\mu$-complex data set $X$ will have low polynomial dependency on $\mu, d, 1/\varepsilon$ but only sublinear dependency on its original size parameter $n$. So for $\mu$-complex data sets having small $\mu(X) \leq \mu$ we have the first $(1 \pm \varepsilon)$-coreset of provably sublinear size. The above definition implies, for $\mu(X) \leq \mu$, the following inequalities. The reader should keep in mind that for all $\beta \in \mathbb{R}^d$

$$\mu^{-1} \|(D_w X\beta)^-\|_1 \leq \|(D_w X\beta)^+\|_1 \leq \mu \|(D_w X\beta)^-\|_1 .$$

We conjecture that computing the value of $\mu(X)$ is hard. However, it can be approximated in polynomial time. It is not necessary to do so in practical applications, but we include this result for those who wish to evaluate whether their data has nice $\mu$-complexity.

**Theorem 3.** *Let $X \in \mathbb{R}^{n \times d}$ be weighted by $w \in \mathbb{R}^n_{>0}$. Then a $\mathrm{poly}(d)$-approximation to the value of $\mu_w(X)$ can be computed in $O(\mathrm{poly}(nd))$ time.*

The parameter $\mu(X)$ has an intuitive interpretation and might be of independent interest. The odds of a binary random variable $V$ are defined as $\frac{\mathbb{P}[V=1]}{\mathbb{P}[V=0]}$. The model assumption of logistic regression is that for every sample $X_i$, the logarithm of the odds is a linear function of $X_i \beta$. For a candidate $\beta$, multiplying all odds and taking the logarithm is then exactly $\|X\beta\|_1$. Our definition now relates the probability mass due to the incorrectly predicted odds and the probability mass due to the correctly

predicted odds. We say that the ratio between these two is upper bounded by $\mu$. For logistic regression, assuming they are within some order of magnitude is not uncommon. One extreme is the (degenerate) case where the data set is exactly separable. Choosing $\beta$ to parameterize a separating hyperplane for which $X\beta$ is all positive, implies that $\mu(X) = \infty$. Another case is when we have a large ratio between the number of positively and negatively labeled points which is a lower bound to $\mu$. Under either of these conditions, logistic regression exhibits methodological weaknesses due to the separation or imbalance between the given classes, cf. [35, 26].

## 3  Lower Bounds

At first glance, one might think of taking a uniform sample as a coreset. We demonstrate and discuss on worst-case instances in Appendix C that this won't work in theory or in practice. In the following we will show a much stronger result, namely that no efficient streaming algorithms or coresets for logistic regression can exist in general, even if we assume that the points lie in 2-dimensional Euclidean space. To this end we will reduce from the INDEX communication game. In its basic variant, there exist two players Alice and Bob. Alice is given a binary bit string $x \in \{0,1\}^n$ and Bob is given an index $i \in [n]$. The goal is to determine the value of $x_i$ with constant probability while using as little communication as possible. Clearly, the difficulty of the problem is inherently one-way; otherwise Bob could simply send his index to Alice. If the entire communication consists of only a single message sent by Alice to Bob, the message must contain $\Omega(n)$ bits [30].

**Theorem 4.** *Let $Z \in \mathbb{R}^{n \times 2}, Y \in \{-1, 1\}^n$ be an instance of logistic regression in 2-dimensional Euclidean space. Any one-pass streaming algorithm that approximates the optimal solution of logistic regression up to any finite multiplicative approximation factor requires $\Omega(n/\log n)$ bits of space.*

A similar reduction also holds if Alice's message consists of points forming a coreset. Hence, the following corollary holds.

**Corollary 5.** *Let $Z \in \mathbb{R}^{n \times 2}, Y \in \{-1, 1\}^n$ be an instance of logistic regression in 2-dimensional Euclidean space. Any coreset of $Z, Y$ for logistic regression consists of at least $\Omega(n/\log n)$ points.*

We note that the proof can be slightly modified to rule out any finite additive error as well. This indicates that the notion of *lightweight* coresets with multiplicative and additive error [4] is not a sufficient relaxation. Independently of our work [41] gave a linear lower bound in a more general context based on a worst case instance to the sensitivity approach due to [27]. Our lower bounds and theirs are incomparable; they show that if a coreset can only consist of input points it comprises the entire data set in the worst-case. We show that no coreset with $o(n/\log n)$ can exist, irrespective of whether input points are used. While the distinction may seem minor, a number of coreset constructions in literature necessitate the use of non-input points, see [1] and [23].

## 4  Sampling via Sensitivity Scores

Our sampling based coreset constructions are obtained with the following approach, called sensitivity sampling. Suppose we are given a data set $X \in \mathbb{R}^{n \times d}$ together with weights $w \in \mathbb{R}^n_{>0}$ as in Definition 1. Recall the function under study is $f_w(X\beta) = \sum_{i=1}^n w_i \cdot g(x_i\beta)$. Associate with each point $x_i$ the function $g_i(\beta) = g(x_i\beta)$. Then we have the following definition.

**Definition 6.** *[31] Consider a family of functions $\mathcal{F} = \{g_1, \ldots, g_n\}$ mapping from $\mathbb{R}^d$ to $[0, \infty)$ and weighted by $w \in \mathbb{R}^n_{>0}$. The sensitivity of $g_i$ for $f_w(\beta) = \sum_{i=1}^n w_i g_i(\beta)$ is*

$$\varsigma_i = \sup \frac{w_i g_i(\beta)}{f_w(\beta)} \tag{1}$$

*where the $\sup$ is over all $\beta \in \mathbb{R}^d$ with $f_w(\beta) > 0$. If this set is empty then $\varsigma_i = 0$. The total sensitivity is $\mathfrak{S} = \sum_{i=1}^n \varsigma_i$.*

The sensitivity of a point measures its worst-case importance for approximating the objective function on the entire input data set. Performing importance sampling proportional to the sensitivities of the input points thus yields a good approximation. Computing the sensitivities is often intractable and involves solving the original optimization problem to near-optimality, which is the problem we want to solve in the first place, as pointed out in [8]. To get around this, it was shown that any upper bound

on the sensitivities $s_i \geq \varsigma_i$ also has provable guarantees. However, the number of samples needed depends on the total sensitivity, that is, the sum of their estimates $S = \sum_{i=1}^{n} s_i \geq \sum_{i=1}^{n} \varsigma_i = \mathfrak{S}$, so we need to carefully control this quantity. Another complexity measure that plays a crucial role in the sampling complexity is the VC dimension of the range space induced by the set of functions under study.

**Definition 7.** *A range space is a pair* $\mathfrak{R} = (\mathcal{F}, \mathrm{ranges})$ *where* $\mathcal{F}$ *is a set and* $\mathrm{ranges}$ *is a family of subsets of* $\mathcal{F}$. *The VC dimension* $\Delta(\mathfrak{R})$ *of* $\mathfrak{R}$ *is the size* $|G|$ *of the largest subset* $G \subseteq \mathcal{F}$ *such that* $G$ *is shattered by* $\mathrm{ranges}$, *i.e.,* $|\{G \cap R \mid R \in \mathrm{ranges}\}| = 2^{|G|}$.

**Definition 8.** *Let* $\mathcal{F}$ *be a finite set of functions mapping from* $\mathbb{R}^d$ *to* $\mathbb{R}_{\geq 0}$. *For every* $\beta \in \mathbb{R}^d$ *and* $r \in \mathbb{R}_{\geq 0}$, *let* $\mathrm{range}_{\mathcal{F}}(\beta, r) = \{f \in \mathcal{F} \mid f(\beta) \geq r\}$, *and* $\mathrm{ranges}(\mathcal{F}) = \{\mathrm{range}_{\mathcal{F}}(\beta, r) \mid \beta \in \mathbb{R}^d, r \in \mathbb{R}_{\geq 0}\}$, *and* $\mathfrak{R}_{\mathcal{F}} = (\mathcal{F}, \mathrm{ranges}(\mathcal{F}))$ *be the range space induced by* $\mathcal{F}$.

Recently a framework combining the sensitivity scores with a theory on the VC dimension of range spaces was developed in [8]. For technical reasons we use a slightly modified version.

**Theorem 9.** *Consider a family of functions* $\mathcal{F} = \{f_1, \dots, f_n\}$ *mapping from* $\mathbb{R}^d$ *to* $[0, \infty)$ *and a vector of weights* $w \in \mathbb{R}_{\geq 0}^n$. *Let* $\varepsilon, \delta \in (0, 1/2)$. *Let* $s_i \geq \varsigma_i$. *Let* $S = \sum_{i=1}^{n} s_i \geq \mathfrak{S}$. *Given* $s_i$ *one can compute in time* $O(|\mathcal{F}|)$ *a set* $R \subset \mathcal{F}$ *of*

$$O\left(\frac{S}{\varepsilon^2}\left(\Delta \log S + \log\left(\frac{1}{\delta}\right)\right)\right)$$

*weighted functions such that with probability* $1 - \delta$ *we have for all* $\beta \in \mathbb{R}^d$ *simultaneously*

$$\left|\sum_{f \in \mathcal{F}} w_i f_i(\beta) - \sum_{f \in R} u_i f_i(\beta)\right| \leq \varepsilon \sum_{f \in \mathcal{F}} w_i f_i(\beta).$$

*where each element of* $R$ *is sampled i.i.d. with probability* $p_j = \frac{s_j}{S}$ *from* $\mathcal{F}$, $u_i = \frac{S w_j}{s_j |R|}$ *denotes the weight of a function* $f_i \in R$ *that corresponds to* $f_j \in \mathcal{F}$, *and where* $\Delta$ *is an upper bound on the VC dimension of the range space* $\mathfrak{R}_{\mathcal{F}^*}$ *induced by* $\mathcal{F}^*$ *that can be obtained by defining* $\mathcal{F}^*$ *to be the set of functions* $f_j \in \mathcal{F}$ *where each function is scaled by* $\frac{S w_j}{s_j |R|}$.

Now we show that the VC dimension of the range space induced by the set of functions studied in logistic regression can be related to the VC dimension of the set of linear classifiers. We first start with a fixed common weight and generalize the result to a more general finite set of distinct weights.

**Lemma 10.** *Let* $X \in \mathbb{R}^{n \times d}, c \in \mathbb{R}_{>0}$. *The range space induced by* $\mathcal{F}_{log}^c = \{c \cdot g(x_i \beta) \mid i \in [n]\}$ *satisfies* $\Delta(\mathfrak{R}_{\mathcal{F}_{log}^c}) \leq d + 1$.

**Lemma 11.** *Let* $X \in \mathbb{R}^{n \times d}$ *be weighted by* $w \in \mathbb{R}^n$ *where* $w_i \in \{v_1, \dots, v_t\}$ *for all* $i \in [n]$. *The range space induced by* $\mathcal{F}_{log} = \{w_i \cdot g(x_i \beta) \mid i \in [n]\}$ *satisfies* $\Delta(\mathfrak{R}_{\mathcal{F}_{log}}) \leq t \cdot (d + 1)$.

We will see later how to bound the number of distinct weights $t$ by a logarithmic term in the range of the involved weights. It remains for us to derive tight and efficiently computable upper bounds on the sensitivities.

**Base Algorithm** We show that sampling proportional to the *square root* of the $\ell_2$-leverage scores augmented by $w_i / \sum_{j \in [n]} w_j$ yields a coreset whose size is roughly linear in $\mu$ and the dependence on the input size is roughly $\sqrt{n}$. In what follows, let $\mathcal{W} = \sum_{i \in [n]} w_i$.

We make a case distinction covered by lemmas 12 and 13. The intuition in the first case is that for a sufficiently large positive entry $z$, we have that $|z| \leq g(z) \leq 2|z|$. The lower bound holds even for all non-negative entries. Moreover, for $\mu$-complex inputs we are able to relate the $\ell_1$ norm of all entries to the positive ones, which will yield the desired bound, arguing similarly to the techniques of [13] though adapted here for logistic regression.

**Lemma 12.** *Let* $X \in \mathbb{R}^{n \times d}$ *weighted by* $w \in \mathbb{R}_{>0}^n$ *be* $\mu$-complex. *Let* $U$ *be an orthonormal basis for the columnspace of* $D_w X$. *If for index* $i$, *the supreme* $\beta$ *in (1) satisfies* $0.5 \leq x_i \beta$ *then* $w_i g(x_i \beta) \leq 2(1 + \mu) \|U_i\|_2 f_w(X \beta)$.

In the second case, the element under study is bounded by a constant. We consider two sub cases. If there are a lot of contributions, which are not too small, and thus cost at least a constant each, then

we can lower bound the total cost by a constant times their total weight. If on the other hand there are many very small negative values, then this implies again that the cost is within a $\mu$ fraction of the total weight.

**Lemma 13.** *Let $X \in \mathbb{R}^{n \times d}$ weighted by $w \in \mathbb{R}^n_{>0}$ be $\mu$-complex. If for index $i$, the supreme $\beta$ in (1) satisfies $0.5 \geq x_i\beta$ then $w_i g(x_i\beta) \leq \frac{(20+\mu)w_i}{\mathcal{W}} f_w(X\beta)$.*

Combining both lemmas yields general upper bounds on the sensitivities that we can use as an importance sampling distribution. We also derive an upper bound on the total sensitivity that will be used to bound the sampling complexity.

**Lemma 14.** *Let $X \in \mathbb{R}^{n \times d}$ weighted by $w \in \mathbb{R}^n_{>0}$ be $\mu$-complex. Let $U$ be an orthonormal basis for the columnspace of $D_wX$. For each $i \in [n]$, the sensitivity of $g_i(\beta) = g(x_i\beta)$ for the weighted logistic regression function is bounded by $\varsigma_i \leq s_i = (20+2\mu)\cdot(\|U_i\|_2 + w_i/\mathcal{W})$. The total sensitivity is bounded by $\mathfrak{S} \leq S \leq 44\mu\sqrt{nd}$.*

We combine the above results into the following theorem.

**Theorem 15.** *Let $X \in \mathbb{R}^{n \times d}$ weighted by $w \in \mathbb{R}^n$ be $\mu$-complex. Let $\omega = \frac{w_{\max}}{w_{\min}}$ be the ratio between the maximum and minimum weight in $w$. Let $\varepsilon \in (0, 1/2)$. There exists a $(1 \pm \varepsilon)$-coreset of $X, w$ for logistic regression of size $k \in O(\frac{\mu\sqrt{n}}{\varepsilon^2}d^{3/2}\log(\mu nd)\log(\omega n))$. Such a coreset can be constructed in two passes over the data, in $O(\mathtt{nnz}(X)\log n + \mathrm{poly}(d)\log n)$ time, and with success probability $1 - 1/n^c$ for any absolute constant $c > 1$.*

**Recursive Algorithm** Here we develop a recursive algorithm, inspired by the recursive sampling technique of [14] for the Huber $M$-estimator, though adapted here for logistic regression. This yields a better dependence on the input size. More specifically, we can diminish the leading $\sqrt{n}$ factor to only $\log^c(n)$ for an absolute constant $c$. One complication is that the parameter $\mu$ grows in the recursion, which we need to control, while another complication is having to deal with the separate $\ell_1$ and uniform parts of our sampling distribution.

We apply the Algorithm of Theorem 15 recursively. To do so, we need to ensure that after one stage of subsampling and reweighting, the resulting data set remains $\mu'$-complex for a value $\mu'$ that is not too much larger than $\mu$. To this end, we first bound the VC dimension of a range space induced by an $\ell_1$ related family of functions.

**Lemma 16.** *The range space induced by $\mathcal{F}_{\ell_1} = \{h_i(\beta) = w_i|x_i\beta| \mid i \in [n]\}$ satisfies $\Delta(\mathfrak{R}_{\mathcal{F}_{\ell_1}}) \leq 10(d+1)$.*

Applying Theorem 9 to $\mathcal{F}_{\ell_1}$ implies that the subsample of Theorem 15 satisfies a so called $\varepsilon$-subspace embedding property for $\ell_1$. Note that, by linearity of the $\ell_1$-norm, we can fold the weights into $D_wX$.

**Lemma 17.** *Let $T$ be a sampling and reweighting matrix according to Theorem 15. I.e., $TD_wX$ is the resulting reweighted sample when Theorem 15 is applied to $\mu$-complex input $X, w$. Then with probability $1 - 1/n^c$, for all $\beta \in \mathbb{R}^d$ simultaneously*
$$(1 - \varepsilon')\|D_wX\beta\|_1 \leq \|TD_wX\beta\|_1 \leq (1 + \varepsilon')\|D_wX\beta\|_1$$
*holds, where $\varepsilon' = \varepsilon/\sqrt{\mu+1}$.*

Using this, we can show that the $\mu$-complexity is not violated too much after one stage of sampling.

**Lemma 18.** *Let $T$ be a sampling and reweighting matrix according to Theorem 15 where parameter $\varepsilon$ is replaced by $\varepsilon/\sqrt{\mu+1}$. That is $TD_wX$ is the resulting reweighted sample when Theorem 15 succeeds on $\mu$-complex input $X, w$. Suppose that simultaneously Lemma 17 holds. Let*
$$\mu' = \mu_{Tw}(X) = \sup_{\beta \in \mathbb{R}^d} \frac{\|(TD_wX\beta)^+\|_1}{\|(TD_wX\beta)^-\|_1}.$$
*Then we have $\mu' \leq (1 + \varepsilon)\mu$.*

Now we are ready to prove our theorem regarding the recursive subsampling algorithm.

**Theorem 19.** *Let $X \in \mathbb{R}^{n \times d}$ be $\mu$-complex. Let $\varepsilon \in (0, 1/2)$. There exists a $(1 \pm \varepsilon)$-coreset of $X$ for logistic regression of size $k \in O(\frac{\mu^3}{\varepsilon^4}d^3\log^2(\mu nd)\log^2 n (\log\log n)^4)$. Such a coreset can be constructed in time $O((\mathtt{nnz}(X) + \mathrm{poly}(d))\log n \log\log n)$ in $2\log(\frac{1}{\eta})$ passes over the data for a small $\eta > 0$, assuming the machine has access to sufficient memory to store and process $\tilde{O}(n^\eta)$ weighted points. The success probability is $1 - 1/n^c$ for any absolute constant $c > 1$.*

# 5   Experiments

We ran a series of experiments to illustrate the performance of our coreset method. All experiments were run on a Linux machine using an Intel i7-6700, 4 core CPU at 3.4 GHz, and 32GB of RAM. We implemented our algorithms in Python. Now, we compare our basic algorithm to simple uniform sampling and to sampling proportional to the sensitivity upper bounds given by [27].

**Implementation Details** The approach of [27] is based on a $k$-means++ clustering [3] on a small uniform sample of the data and was performed using standard parameters taken from the publication. For this purpose we used parts of their original Python code. However, we removed the restriction of the domain of optimization to a region of small radius around the origin. This way, we enabled unconstrained regression in the domain $\mathbb{R}^d$. The exact QR-decomposition is rather slow on large data matrices. We thus optimized the running time of our approach in the following way. We used a fast approximation algorithm based on the sketching techniques of [12], cf. [43]. That leads to a provable constant approximation of the square root of the leverage scores with constant probability, cf. [18], which means that the total sensitivity bounds given in our theory will grow by only a small constant factor. A detailed description of the algorithm is in the proof of Theorem 15. The subsequent optimization was done for all approaches with the standard gradient based optimizer from the `scipy.optimize` package, see `http://www.scipy.org/`.

**Data Sets** We briefly introduce the data sets that we used. The WEBB SPAM[1]  data consists of $350,000$ unigrams with $127$ features from web pages which have to be classified as spam or normal pages ($61\%$ positive). The COVERTYPE[2]  data consists of $581,012$ cartographic observations of different forests with $54$ features. The task is to predict the type of trees at each location ($49\%$ positive). The KDD CUP '99[3]  data comprises $494,021$ network connections with $41$ features and the task is to detect network intrusions ($20\%$ positive).

**Experimental Assessment** For each data set we assessed the total running times for computing the sampling probabilities, sampling and optimizing on the sample. In order to assess the approximation accuracy we examined the relative error $|\mathcal{L}(\beta^*|X) - \mathcal{L}(\tilde{\beta}|X)|/\mathcal{L}(\beta^*|X)$ of the negative log-likelihood for the maximum likelihood estimators obtained from the full data set $\beta^*$ and the subsamples $\tilde{\beta}$. For each data set, we ran all three subsampling algorithms for a number of thirty regular subsampling steps in the range $k \in [\lfloor 2\sqrt{n} \rfloor, \lceil n/16 \rceil]$. For each step, we present the mean relative error as well as the trade-off between mean relative error and running time, taken over twenty independent repetitions, in Figure 1. Relative running times, standard deviations and absolute values are presented in Figure 2 respectively in Table 1 in Appendix B.

**Evaluation** The accuracy of the QR-sampling distribution outperforms uniform sampling and the distribution derived from $k$-means on all instances. This is especially true for small sampling sizes. Here, the relative error especially for uniform sampling tends to deteriorate. While $k$-means sampling occasionally improved over uniform sampling for small sample sizes, the behavior of both distributions was similar for larger sampling sizes. The standard deviations had a similarly low magnitude as the mean values, where the QR method usually showed the lowest values. The trade-off between the running time and relative errors shows a common picture for WEBB SPAM and COVERTYPE. QR is nearly always more accurate than the other algorithms for a similar time budget, except for regions where the relative error is large, say above 5-10% while for larger time budgets, QR is better by a factor between 1.5-3 and drops more quickly towards 0. The conclusion so far could be that for a quick guess, say a 1.1-approximation, the competitors are faster, but to provably obtain a reasonably small relative error below 5%, QR outperforms its competitors. However, for KDD CUP '99, QR always has a lower error than its competitors. Their relative errors remain above 15% or much worse, while QR never exceeds 22% and drops quickly below 4%. Our estimates for $\mu$ support that KDD CUP '99 seems more difficult to approximate than the others. The estimates were 4.39 for WEBB SPAM, 1.86 for COVERTYPE, and 35.18 for KDD CUP '99.

The relative running time for the QR-distribution was comparable to $k$-means and only slightly higher than uniform sampling. However, it never exceeded a factor of two compared to its competitors and remained negligible compared to the full optimization task, see Figure 2 in Appendix B. The standard deviations were negligible except for the $k$-means algorithm and the KDD CUP '99 data

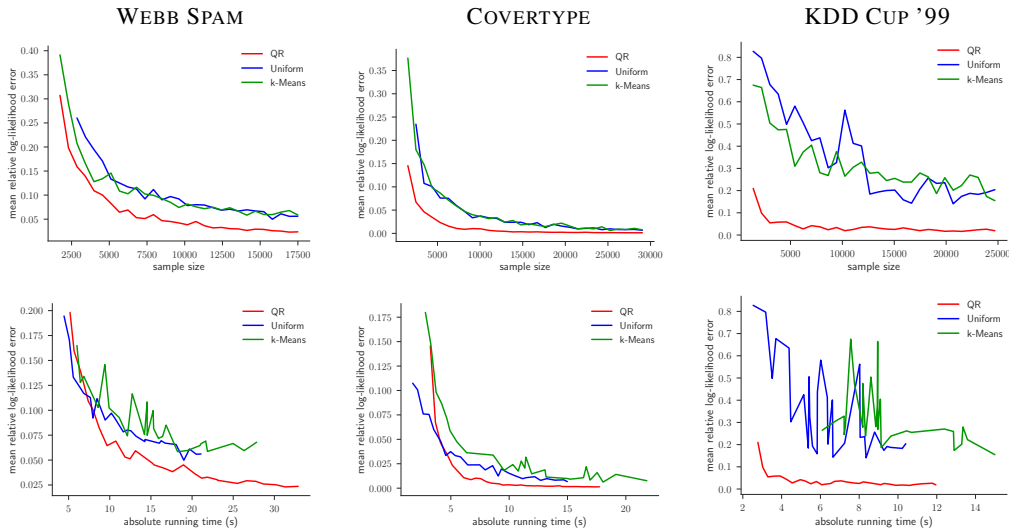

Figure 1: Each column shows the results for one data set comprising thirty different coreset sizes (depending on the individual size of the data sets). The plotted values are means taken over twenty independent repetitions of each experiment. The plots in the upper row show the mean relative log-likelihood errors of the three subsampling distributions, uniform sampling (blue), our QR derived distribution (red), and the $k$-means based distribution (green). All values are relative to the corresponding optimal log-likelihood values of the optimization task on the full data set. The plots in the lower row show the trade-off between running time and relative errors (lower is better).

set, where the uniform and $k$-means based algorithms showed larger values. The QR method had much lower standard deviations. This indicates that the resulting coresets are more stable for the subsequent numerical optimization. We note that the savings of all presented data reduction methods become even more significant when performing more time consuming data analysis tasks like MCMC sampling in a Bayesian setting, see e.g., [27, 24].

## 6 Conclusions

We first showed that (sublinear) coresets for logistic regression do not exist in general. It is thus necessary to make further assumptions on the nature of the data. To this end we introduced a new complexity measure $\mu(X)$, which quantifies the amount of overlap of positive and negative classes and the balance in their cardinalities. We developed the first rigorously sublinear $(1 \pm \varepsilon)$-coresets for logistic regression, given that the original data has small $\mu$-complexity. The leading factor is $O(\varepsilon^{-2}\mu\sqrt{n})$. We have further developed a recursive coreset construction that reduces the dependence on the input size to only $O(\log^c n)$ for absolute constant $c$. This comes at the cost of an increased dependence on $\mu$. However, it is beneficial for very large and well-behaved data. Our algorithms are space efficient, and can be implemented in a variety of models, used to tackle the challenges of large data sets, such as 2-pass streaming, and massively parallel frameworks like Hadoop and MapReduce, and can be implemented to run in input sparsity time $\tilde{O}(\mathtt{nnz}(X))$, which is especially beneficial for sparsely encoded input data. Our experimental evaluation shows that our implementation of the basic algorithm outperforms uniform sampling as well as state of the art methods in the area of coresets for logistic regression while being competitive to both regarding its running time.

## Acknowledgments

We thank the anonymous reviewers for their valuable comments. We also thank our assistant Moritz Paweletz. This work was supported by the German Science Foundation (DFG) Collaborative Research Center SFB 876, projects A2 and C4. Chris Schwiegelshohn is supported in part by an ERC Advanced Grant 788893 AMDROMA. David P. Woodruff is supported in part by an Office of Naval Research (ONR) grant N00014-18-1-2562, and part of this work was done while he was visiting the Simons Institute for the Theory of Computing.

## Footnotes

[1]`https://www.cc.gatech.edu/projects/doi/WebbSpamCorpus.html`

[2]`https://archive.ics.uci.edu/ml/datasets/covertype`

[3]`http://kdd.ics.uci.edu/databases/kddcup99/kddcup99.html`

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
