[Supplementary Material]

# A  Proofs

*Proof of Theorem 3.* Let $A = D_w X$. In $O(nd \log d + \mathrm{poly}(d))$ time, find an $\ell_1$-well-conditioned basis [11] $U \in \mathbb{R}^{n \times d}$ of $A$, such that

$$\forall \beta \in \mathbb{R}^d \colon \|\beta\|_1 \leq \|U\beta\|_1 \leq \mathrm{poly}(d)\|\beta\|_1.$$

Then $\mu(U)$ and $\mu(A)$ are the same since $U$ and $A$ span the same columnspace. By linearity it suffices to optimize over unit-$\ell_1$ vectors $\beta$. If we minimize $\|(U\beta)^-\|_1$ over unit-$\ell_1$ vectors $\beta$, and $t$ is the minimum value, then $\mu$ is at most $\mathrm{poly}(d)/t$, and at least $1/t$ by the well-conditioned basis property, so we just need to find $t$, which can be done with the following linear program:

$$
\begin{aligned}
\min \quad & \sum_{i=1}^{n} b_i \\
\text{s.t.} \quad & \forall i \in [n] \colon (U\beta)_i = a_i - b_i \\
& \forall i \in [d] \colon \beta_i = c_i - d_i \\
& \sum_{i=1}^{d} c_i + d_i \geq 1 \\
& \forall i \in [n] \colon a_i, b_i \geq 0 \\
& \forall i \in [d] \colon c_i, d_i \geq 0
\end{aligned}
$$

Note that $\sum_{i=1}^{d} c_i + d_i \geq 1$ ensures $\|\beta\|_1 \geq 1$, but to minimize the objective function, one will always have $\|\beta\|_1$. Further, if both $a_i$ and $b_i$ are positive for some $i$, they can both be reduced, reducing the objective function. So $\sum_{i=1}^{n} b_i$ exactly corresponds to the minimum over $\beta \in \mathbb{R}^d$ of $\|(U\beta)^-\|_1$. $\qquad\square$

*Proof of Theorem 4.* Assume we had a streaming algorithm using $o(n/\log n)$ space. We construct the following protocol for INDEX: Consider an instance of INDEX, i.e., Alice has a string $x \in \{0,1\}^n$ and Bob has an index $i \in [n]$. We transform the instance into an instance for logistic regression. For each $x_j = 1$, Alice adds a point $p_j = (\cos(\frac{j}{n}), \sin(\frac{j}{n}))$. Note that all of these points have unit Euclidean norm and hence any single point may be linearly separated from the others. All of Alice's points have label 1. Alice summarizes the point set by running the streaming algorithm and sends a message containing the working memory of the streaming algorithm to Bob. Bob now adds the point $p_i = (1-\delta) \cdot (\cos(\frac{i}{n}), \sin(\frac{i}{n}))$ for small enough $\delta > 0$ with label $-1$. From the contents of Alice's message and $p_i$, Bob now obtains a solution to the logistic regression instance. Clearly, if Alice added $p_i$ and hence $x_i = 1$ then the optimal solution will have cost at least $\ln(2)$, since there will be at least one misclassification. If, on the other hand, Alice did not add $p_i$ and hence $x_i = 0$, then the two point sets are linearly separable and the cost tends to 0. Distinguishing between these two cases, i.e. approximating the cost of logistic regression beyond a factor $\lim_{x \to 0} \frac{\ln(2)}{x}$ solves the INDEX problem.

To conclude the theorem, let us consider the space required to encode the points added by Alice. For the reduction to work, it is only important that any point added by Alice can be linearly separated from the others. This can be achieved by using $O(\log n)$ bits per point, i.e., the space of Alice's point set is at most $n' \in O(n \log n)$. The space bound now follows from the lower bound of $\Omega(n) \subseteq \Omega(n'/\log n)$ bits due to [30] for the INDEX problem. $\qquad\square$

*Proof of Corollary 5.* If we had a coreset construction with $o(n/\log n)$ points, we have a protocol for INDEX: Alice computes a coreset for her point set defined in the proof of Theorem 4 and sends it to Bob. Bob computes an optimal solution on the union of the coreset and his point. This solves INDEX using $o(n)$ communication, which contradicts the lower bound of [30]. So Alice's coreset cannot exist. $\qquad\square$

*Proof of Lemma 10.* (cf. [27]) For all $G \subseteq \mathcal{F}_{log}^c$, we have

$$|\{G \cap R \mid R \in \mathrm{ranges}(\mathcal{F}_{log}^c)\}| = |\{\mathrm{range}_G(\beta, r) \mid \beta \in \mathbb{R}^d, r \in \mathbb{R}_{\geq 0}\}|$$

Note that $g$ is invertible and monotone. Also note that $g^{-1}$ maps $\mathbb{R}_{\geq 0}$ surjectively into $\mathbb{R}$. For all $\beta \in \mathbb{R}^d, r \in \mathbb{R}_{\geq 0}$ we thus have

$$\mathrm{range}_G(\beta, r) = \{c \cdot g_i \in G \mid c \cdot g_i(\beta) \geq r\}$$

$$= \{c \cdot g_i \in G \mid c \cdot g(x_i\beta) \geq r\} = \{c \cdot g_i \in G \mid x_i\beta \geq g^{-1}(r/c)\}.$$

Now note that $\{c \cdot g_i \in G \mid x_i\beta \geq g^{-1}(r/c)\}$ corresponds to the set of points that is shattered by the affine hyperplane classifier $x_i \mapsto \mathbf{1}_{\{x_i\beta - g^{-1}(r/c) \geq 0\}}$. We can conclude that

$$\left| \{\text{range}_G(\beta, r) \mid \beta \in \mathbb{R}^d, r \in \mathbb{R}_{\geq 0}\} \right| = \left| \{\{g_i \in G \mid x_i\beta - s \geq 0\} \mid \beta \in \mathbb{R}^d, s \in \mathbb{R}\} \right|$$

which means that the VC dimension of $\mathfrak{R}_{\mathcal{F}_{log}^c}$ is $d+1$ since the VC dimension of the set of hyperplane classifiers is $d+1$ [29, 42]. $\qquad\square$

*Proof of Lemma 11.* We partition the functions into $t$ disjoint classes having equal weights. Let $F_i = \{w_j \cdot g_j \in \mathcal{F}_{log} \mid w_j = v_i\}$, for $i \in [t]$. For the sake of contradiction, suppose $\Delta(\mathfrak{R}_{\mathcal{F}_{log}}) > t \cdot (d+1)$. Then there exists a set $G$ of size $|G| > t \cdot (d+1)$ that is shattered by the ranges of $\mathfrak{R}_{\mathcal{F}_{log}}$. Now consider the sets $F_i \cap G$, for $i \in [t]$. Due to the disjointness property, each set $F_i \cap G$ must be shattered by the ranges induced by $F_i$. But at least one of them must be as large as $\frac{|G|}{t} > \frac{t\cdot(d+1)}{t} = d+1$, which contradicts Lemma 10. Thus $\Delta(\mathfrak{R}_{\mathcal{F}_{log}}) \leq t \cdot (d+1) \in O(dt)$ follows. $\qquad\square$

*Proof of Lemma 12.* Let $D_w X = UR$, where $U$ is an orthonormal basis for the columnspace of $D_w X$. It follows from $0.5 \leq x_i\beta$ and monotonicity of $g$ that

$$w_i g(x_i\beta) = w_i g\left(\frac{w_i x_i\beta}{w_i}\right) = w_i g\left(\frac{U_i R\beta}{w_i}\right) \leq w_i g\left(\frac{\|U_i\|_2 \|R\beta\|_2}{w_i}\right) = w_i g\left(\frac{\|U_i\|_2 \|UR\beta\|_2}{w_i}\right)$$

$$= w_i g\left(\frac{\|U_i\|_2 \|D_w X\beta\|_2}{w_i}\right) \leq w_i \frac{2}{w_i} \|U_i\|_2 \|D_w X\beta\|_2 \leq 2\|U_i\|_2 \|D_w X\beta\|_1$$

$$\leq 2\|U_i\|_2(1+\mu)\|(D_w X\beta)^+\|_1 = 2\|U_i\|_2(1+\mu) \sum_{j: w_j x_j\beta \geq 0} w_j |x_j\beta|$$

$$\leq 2\|U_i\|_2(1+\mu) \sum_{j: x_j\beta \geq 0} w_j g(x_j\beta) \leq 2\|U_i\|_2(1+\mu) f_w(X\beta). \qquad\square$$

*Proof of Lemma 13.* Let $K^- = \{j \in [n] \mid x_j\beta \leq -2\}$ and $K^+ = \{j \in [n] \mid x_j\beta > -2\}$. Note that $g(-2) > 1/10$ and $g(x_i\beta) \leq g(0.5) < 1$. Also, $\sum_{j \in K^-} w_j + \sum_{j \in K^+} w_j = \mathcal{W}$.

Thus if $\sum_{j \in K^+} w_j \geq \frac{1}{2}\mathcal{W}$ then

$$f_w(X\beta) = \sum_{i=1}^{n} w_j g(x_j\beta) \geq \frac{\sum_{j \in [n]} w_j}{20} \geq \frac{\mathcal{W}}{20 w_i} \cdot w_i g(x_i\beta).$$

If on the other hand $\sum_{j \in K^+} w_j < \frac{1}{2}\mathcal{W}$ then $\sum_{j \in K^-} w_j \geq \frac{1}{2}\mathcal{W}$. Thus

$$f_w(X\beta) \geq \|(D_w X\beta)^+\|_1 \geq \|(D_w X\beta)^-\|_1/\mu \geq \left(2 \cdot \frac{\sum_{j \in [n]} w_j}{2}\right) \Big/ \mu \geq \frac{\mathcal{W}}{\mu w_i} \cdot w_i g(x_i\beta). \square$$

*Proof of Lemma 14.* From Lemma 12 and Lemma 13 we have for each $i$

$$\varsigma_i = \sup_\beta \frac{w_i g(x_i\beta)}{f_w(X\beta)} \leq 2(1+\mu)\|U_i\|_2 + (20+\mu)\frac{w_i}{\mathcal{W}} \leq (20+2\mu)\left(\|U_i\|_2 + \frac{w_i}{\mathcal{W}}\right)$$

From this, the second claim follows via the Cauchy-Schwarz inequality and using the fact that the Frobenius norm satisfies $\|U\|_F = \sqrt{\sum_{i \in [n], j \in [d]} |U_{ij}|^2} = \sqrt{d}$ due to orthonormality of $U$. We have

$$\mathfrak{S} = \sum_{i=1}^{n} \varsigma_i \leq (20+2\mu)\sum_{i=1}^{n}\left(\|U_i\|_2 + \frac{w_i}{\mathcal{W}}\right) \leq 22\mu(\sqrt{n}\|U\|_F + 1) \leq 44\mu\sqrt{n}d. \qquad\square$$

*Proof of Theorem 15.* The algorithm computes the QR-decomposition $D_w X = QR$ of $D_w X$. Note that $Q$ is an orthonormal basis for the columnspace of $D_w X$. We would like to use the upper bounds on the sensitivities from Lemma 14. Namely, to sample the input points proportional to the sampling probabilities $\frac{s_i}{\sum_{j=1}^{n} s_j} = \frac{\|Q_i\|_2 + w_i/\mathcal{W}}{\sum_{j=1}^{n}(\|Q_j\|_2 + w_j/\mathcal{W})}$. However, to keep control of the VC dimension of

the involved range space, we modify them to obtain upper bounds $s_i'$ such that each value $s_i'/w_i$ corresponds to $s_i/w_i$ but is rounded up to the closest power of two. It thus holds $s_i \leq s_i' \leq 2s_i$ for all $i \in [n]$. The input points are sampled proportional to the sampling probabilities $p_i = s_i'/\sum_{j=1}^{n} s_j'$. From Lemma 14 we know that $S' = \sum_{j=1}^{n} s_j' \leq 2S \in O(\mu\sqrt{nd})$.

In the proof of Theorem 9, the VC dimension bound is applied to a set of functions which are reweighted by $\frac{S'w_i}{s_i'k}$. We denote this set of functions $\mathcal{F}_{log}$. Now note that the sensitivities satisfy

$$\frac{2}{w_{\min}} \geq \frac{2}{w_i} \geq \frac{s_i'}{w_i} \geq \frac{s_i}{w_i} \geq \sup_{\beta} \frac{g(x_i\beta)}{\sum_{j=1}^{n} w_j g(x_j\beta)} \overset{\beta=0}{\geq} \frac{1}{\sum_{j=1}^{n} w_j} \geq \frac{1}{nw_{\max}} . \tag{2}$$

Also note that $k$ and $S'$ are fixed values. Since the values $s_i'/w_i$ are scaled to powers of two, by (2) there can be at most $O(\log \frac{nw_{\max}}{w_{\min}}) \subseteq O(\log(\omega n))$ distinct values of $\frac{S'w_i}{s_i'k}$. Putting this into Lemma 11, we have $\Delta(\mathfrak{R}_{\mathcal{F}_{log}}) \in O(d\log(\omega n))$.

Putting all these pieces into Theorem 9 for error parameter $\varepsilon \in (0, 1/2)$ and failure probability $\eta = n^{-c}$, we have that a reweighted random sample of size

$$k \in O\left(\frac{S'}{\varepsilon^2}\left(\Delta(\mathfrak{R}_{\mathcal{F}_{log}})\log S' + \log\left(\frac{1}{\eta}\right)\right)\right)$$

$$\subseteq O\left(\frac{\mu\sqrt{nd}}{\varepsilon^2}\left(d\log(\mu\sqrt{nd})\log(\omega n) + \log(n^c)\right)\right)$$

$$\subseteq O\left(\frac{\mu\sqrt{n}}{\varepsilon^2}d^{3/2}\log(\mu nd)\log(\omega n)\right)$$

is a $(1 \pm \varepsilon)$ coreset with probability $1 - 1/n^c$ as claimed.

It remains to prove the claims regarding streaming and running time. We can compute the QR-decomposition of $D_w X$ in time $O(nd^2)$, see [25]. Once $Q$ is available, we can inspect it row-by-row computing $\|Q_i\|_2 + w_i/\mathcal{W}$ and give it as input together with $x_i$ to $k$ independent copies of a weighted reservoir sampler [9], which takes $O(\text{nnz}(X))$ time to collect all sampled non-zero entries. This gives a total running time of $O(nd^2)$ since the computations are dominated by the QR-decomposition.

We argue how to implement the first step in one streaming pass over the data in time $O(\text{nnz}(X)\log n + \text{poly}(d))$. Using the sketching techniques of [12], cf. [43], we can obtain a provably constant approximation of the square root of the leverage scores $\|Q_i\|_2$ with constant probability [18]. This means that the total sensitivity bound $S$ grows only by a small constant factor and does not affect the asymptotic analysis presented above. The idea is to first sketch the data matrix $X \in \mathbb{R}^{n \times d}$ to a significantly smaller matrix $\tilde{X} \in \mathbb{R}^{n' \times d}$, where $n' \in O(d^2)$. This takes only $O(\text{nnz}(X)\log n + \text{poly}(d)\log n)$ time, where the $\text{poly}(d)$ and $\log n$ factors are only needed to amplify the success probability from constant to $\frac{1}{n^c}$ [43]. Performing the QR-decomposition $\tilde{X} = \tilde{Q}\tilde{R}$ takes $O(n'd^2) \subseteq O(d^4)$ time.

Now, to compute a fast approximation to the row norms, we use a Johnson-Lindenstrauss transform, i.e., a matrix $G \in \mathbb{R}^{d \times m}, m \in O(\log n)$, whose entries are i.i.d. $G_{ij} \sim N(0, \frac{1}{m})$ [28]. We compute the approximation to the row norms used in our sampling probabilities in a second pass over the data, as $\|\tilde{U}_i\|_2 = \|X_i(\tilde{R}^{-1}G)\|_2$, for $i \in [n]$. As we do so, we can feed these augmented with the corresponding weight directly to the reservoir sampler. The latter is a streaming algorithm itself and updates its sample in constant time. The matrix product $\tilde{R}^{-1}G$ takes at most $O(d^2\log n)$ time, and the streaming pass can be done in $O(\text{nnz}(X)\log n)$.

This sums up to two passes over the data and a running time of $O(\text{nnz}(X)\log n + \text{poly}(d)\log n)$. $\quad\square$

*Proof of Lemma 16.* Fix an arbitrary $G \subseteq \mathcal{F}_{\ell_1}$. Let $\Omega = \mathbb{R}^d \times \mathbb{R}_{\geq 0}$. We attempt to bound the quantity

$$|\{G \cap R \mid R \in \text{ranges}(\mathcal{F}_{\ell_1})\}|$$

$$= |\{\text{range}_G(\beta, r) \mid \beta \in \mathbb{R}^d, r \in \mathbb{R}_{\geq 0}\}|$$

$$= \left| \bigcup_{(\beta,r)\in\Omega} \{\{h_i \in G \mid h_i(\beta) \geq r\}\} \right|$$

$$= \left| \bigcup_{(\beta,r)\in\Omega} \{\{h_i \in G \mid w_i x_i \beta \geq r \vee -w_i x_i \beta \geq r\}\} \right|$$

$$\leq \left| \bigcup_{(\beta,r)\in\Omega} \{\{h_i \in G \mid w_i x_i \beta \geq r\}\} \right| \cdot \left| \bigcup_{(\beta,r)\in\Omega} \{\{h_i \in G \mid -w_i x_i \beta \geq r\}\} \right|$$

$$= \left| \bigcup_{(\beta,r)\in\Omega} \{\{h_i \in G \mid w_i x_i \beta \geq r\}\} \right|^2 . \tag{3}$$

The inequality holds, since each non-empty set in the collection on the LHS satisfies either of the conditions of the sets in the collections on the RHS, or both, and is thus the union of two of those sets, one from each collection. It can thus comprise at most all unions obtained from combining any two of these sets. The last equality holds since for each fixed $\beta$ we also union over $-\beta$ as we reach over all $\beta \in \mathbb{R}^d$. The two sets are thus equal.

Now note that each set $\{h_i \in G \mid w_i x_i \beta \geq r\}$ equals the set of weighted points that is shattered by the affine hyperplane classifier $w_i x_i \mapsto \mathbf{1}_{\{w_i x_i \beta - r \geq 0\}}$. Note that the VC dimension of the set of hyperplane classifiers is $d + 1$ [29, 42]. To conclude the claimed bound on $\Delta(\mathfrak{R}_{\mathcal{F}_{\ell_1}})$ it is sufficient to show that the above term (3) is bounded strictly below $2^{|G|}$ for $|G| = 10(d + 1)$. By a bound given in [7, 29] we have for this particular choice

$$(3) \leq \left| \{\{h_i \in G \mid w_i x_i \beta - r \geq 0\} \mid \beta \in \mathbb{R}^d, r \in \mathbb{R}\} \right|^2 \leq \left( \frac{e|G|}{d+1} \right)^{2(d+1)}$$

$$< 2^{2(d+1)\log(30)} \leq 2^{2(d+1)5} = 2^{|G|}$$

which implies that $\Delta(\mathfrak{R}_{\mathcal{F}_{\ell_1}}) < 10(d + 1)$. $\qquad\square$

*Proof of Lemma 17.* Consider any $\beta \in \mathbb{R}^d$. Let $D_w X = UR$ where $U$ is an orthonormal basis for the columnspace of $D_w X$. As in Lemma 12 we have for each index $i$

$$|w_i x_i \beta| = |U_i R\beta| \leq \|U_i\|_2 \|R\beta\|_2 = \|U_i\|_2 \|D_w X\beta\|_2 \leq \|U_i\|_2 \|D_w X\beta\|_1 \tag{4}$$

The sensitivity for the $\ell_1$ norm function of $x_i \beta$ is thus

$$\sup_{\beta\in\mathbb{R}^d\setminus\{0\}} \frac{w_i|x_i\beta|}{\|D_w X\beta\|_1} \leq \|U_i\|_2.$$

Note that our upper bounds on the sensitivities satisfy $s_i \geq \|U_i\|_2$. Thus also $S = \sum_{i=1}^n s_i \geq \sum_{i=1}^n \|U_i\|_2$ holds. In particular, these values are exceeded by more than a factor of $\mu + 1$. Also, by Lemma 16, we have a bound of $O(d)$ on the VC dimension of the class of functions $\mathcal{F}_{\ell_1}$. Now, rescaling the error probability parameter $\delta$ that we put into Theorem 9 by a factor of $\frac{1}{2}$, and union bound over the two sets of functions $\mathcal{F}_{log}$, and $\mathcal{F}_{\ell_1}$, the sample in Theorem 15 satisfies at the same time the claims of Theorem 15 with parameter $\varepsilon$ and of this lemma with parameter $\varepsilon' \leq \varepsilon/\sqrt{\mu+1}$ by folding the additional factor of $\mu + 1$ into $\varepsilon$. $\qquad\square$

*Proof of Lemma 18.* For brevity of presentation let $X' = D_w X$. First note that combining the choice of the parameter $\varepsilon/\sqrt{\mu+1}$ with Lemma 17 we have for all $\beta \in \mathbb{R}^d$

$$(1 - \varepsilon') \|X'\beta\|_1 \leq \|TX'\beta\|_1 \leq (1 + \varepsilon') \|X'\beta\|_1,$$

where $\varepsilon' \leq \frac{\varepsilon}{\mu+1}$. Note that since the weights are non-negative, sampling and reweighting does not change the sign of the entries. This implies for $\eta^+ = |\|(TX'\beta)^+\|_1 - \|(X'\beta)^+\|_1|$ and $\eta^- = |\|(TX'\beta)^-\|_1 - \|(X'\beta)^-\|_1|$ that $\max\{\eta^+, \eta^-\} \leq \eta^+ + \eta^- = |\|TX'\beta\|_1 - \|X'\beta\|_1| \leq \varepsilon'\|X'\beta\|_1$.
From this and $\|X'\beta\|_1 = \|(X'\beta)^+\|_1 + \|(X'\beta)^-\|_1 \leq (\mu + 1)\min\{\|(X'\beta)^+\|_1, \|(X'\beta)^-\|_1\}$ it follows for any $\beta \in \mathbb{R}^d$

$$\frac{\|(TX'\beta)^+\|_1}{\|(TX'\beta)^-\|_1} \leq \frac{\|(X'\beta)^+\|_1 + \varepsilon'\|X'\beta\|_1}{\|(X'\beta)^-\|_1 - \varepsilon'\|X'\beta\|_1} \leq \frac{\|(X'\beta)^+\|_1 + \varepsilon'(\mu+1)\|(X'\beta)^+\|_1}{\|(X'\beta)^-\|_1 - \varepsilon'(\mu+1)\|(X'\beta)^-\|_1}$$

$$\leq \frac{\|(X'\beta)^+\|_1(1+\varepsilon)}{\|(X'\beta)^-\|_1(1-\varepsilon)} \leq \mu \frac{1+\varepsilon}{1-\varepsilon} \leq (1+4\varepsilon)\mu.$$

The claim follows by folding the constant $\frac{1}{4}$ into $\varepsilon$. $\qquad\square$

*Proof of Theorem 19.* Recall, due to Lemma 18, the $\mu'$-complexity at the $i$-th recursion level is upper bounded by $\mu(1+\varepsilon)^i$. We thus apply Theorem 15 recursively $l = \log\log n$ times with parameter $\varepsilon_i = \frac{\varepsilon}{2l\sqrt{\mu+1}(1+\varepsilon)^i}$ for $i \in \{0 \ldots l-1\}$. First we bound the approximation ratio, which is the product of the single stages. We have

$$\prod_{i=0}^{l-1}(1+\varepsilon_i) \leq \prod_{i=0}^{l-1}\left(1+\frac{\varepsilon}{(1+\varepsilon)^i 2l\sqrt{\mu}}\right) \leq \left(1+\frac{\varepsilon}{2l\sqrt{\mu}}\right)^l \leq \exp\left(\frac{\varepsilon}{2\sqrt{\mu}}\right) \leq 1+\frac{\varepsilon}{\sqrt{\mu}}.$$

Also

$$\prod_{i=0}^{l-1}(1-\varepsilon_i) \geq \prod_{i=0}^{l-1}\left(1-\frac{\varepsilon}{(1+\varepsilon)^i 2l\sqrt{\mu}}\right) \geq \prod_{i=0}^{l-1}\left(1-\frac{\varepsilon}{2l\sqrt{\mu}}\right) \geq 1-\sum_{i=0}^{l-1}\frac{\varepsilon}{2l\sqrt{\mu}} \geq 1-\frac{\varepsilon}{2\sqrt{\mu}}.$$

Initially all weights are equal to one. So in the first application of Theorem 15 we have $\omega = 1$. This value might grow as the weights are reassigned. However, from Inequality (2) and the discussion below it follows, that the value of $\omega$ can grow only by a factor of $2n$ in each recursive iteration. So it remains bounded by $\omega \leq (2n)^{\log\log n}$ in all levels of our recursion. Its contribution to the lower order terms given in Theorem 15 is thus bounded by $O\left(\log((2n)^{1+\log\log n})\right) \subseteq O\left(\log n \log\log n\right)$.

The size of the data set at recursion level $i+1$ satisfies

$$n_{i+1} \leq \sqrt{n_i} \cdot \frac{Cl^2(1+\varepsilon)^{2i}\mu^2}{\varepsilon^2}d^{3/2}\log((1+\varepsilon)^i\mu nd)\log n \log\log n$$

$$\leq \sqrt{n_i} \cdot \frac{Cl^2 4^i \mu^2}{\varepsilon^2}d^{3/2}\log(2^i\mu nd)\log n \log\log n$$

for some constant $C > 1$. Solving the recursion until we reach $n_0 = n$ we get the following bound on $n_l$. We use that for our choice $l = \log\log n$ we have $2^l = \log n$ and $n^{2^{-l}} = 2^{\frac{\log n}{2^l}} = 2$.

$$n_l \leq n^{2^{-l}}\prod_{i=0}^{l}\left(C \cdot \frac{l^2 4^i \mu^2}{\varepsilon^2}d^{3/2}\log(2^i\mu nd)\log n \log\log n\right)^{\frac{1}{2^i}}$$

$$\leq 2\prod_{i=0}^{l}4^{\frac{i}{2^i}}\prod_{i=0}^{l}\left(C \cdot \frac{l^2\mu^2}{\varepsilon^2}d^{3/2}\log(2^l\mu nd)\log n \log\log n\right)^{\frac{1}{2^i}}$$

$$\leq 2\prod_{i=0}^{l}4^{\frac{i}{2^i}}\prod_{i=0}^{l}\left(2C \cdot \frac{l^2\mu^2}{\varepsilon^2}d^{3/2}\log(\mu nd)\log n \log\log n\right)^{\frac{1}{2^i}}$$

$$\leq 2 \cdot 4^{\sum_{i=0}^{l}\frac{i}{2^i}}\left(2C \cdot \frac{l^2\mu^2}{\varepsilon^2}d^{3/2}\log(\mu nd)\log n \log\log n\right)^{\sum_{i=0}^{l}\frac{1}{2^i}}$$

$$\leq 2 \cdot 4^2\left(2C \cdot \frac{l^2\mu^2}{\varepsilon^2}d^{3/2}\log(\mu nd)\log n \log\log n\right)^2$$

$$\leq 2 \cdot 16 \cdot 4C^2 \cdot \frac{l^4\mu^4}{\varepsilon^4}d^3\log^2(\mu nd)\log^2 n (\log\log n)^2$$

We conclude that for some constant $C' > C$

$$n_l \leq C' \cdot \frac{l^4\mu^4}{\varepsilon^4}d^3\log^2(\mu nd)\log^2 n (\log\log n)^2 \leq C' \cdot \frac{\mu^4}{\varepsilon^4}d^3\log^2(\mu nd)\log^2 n (\log\log n)^6.$$

To reduce this even further, note that in the final iteration we do not need to preserve the $\mu$-complexity. We can thus apply Theorem 15 with the original approximation parameter $\varepsilon$ to obtain a coreset as claimed of size

$$k \in O\left(\sqrt{n_l} \cdot \frac{\mu}{\varepsilon^2}d^{3/2}\log(\mu nd)\log(\omega n)\right)$$

$$\subseteq O\left(\frac{\mu^2}{\varepsilon^2}d^{3/2}\log(\mu nd)\log n\,(\log\log n)^3 \cdot \frac{\mu}{\varepsilon^2}d^{3/2}\log(\mu nd)\log n\,(\log\log n)\right)$$

$$\subseteq O\left(\frac{\mu^3}{\varepsilon^4}d^3\log^2(\mu nd)\log^2 n\,(\log\log n)^4\right).$$

It remains to bound the failure probability. Note that we use a $\log n$ factor in the sampling sizes at all stages rather than $\log n_i$. The failure probability at each stage is thus bounded by $\frac{1}{n^{c'}}$ for $c' = c + 1 > 2$ by adjusting constants. We can thus take a union bound over the stages to get an error probability of at most

$$l \cdot \frac{1}{n^{c'}} = \frac{\log\log n}{n^{c'}} \leq \frac{1}{n^{c'-1}} \leq \frac{1}{n^c}.$$

Now recall from Theorem 15 the two pass streaming algorithm whose running time was dominated by $O(\mathtt{nnz}(X)\log n_i + \mathrm{poly}(d)\log n_i)$. We can thus bound the running time of the recursive algorithm for sufficiently large $C > 1$ by

$$C(\mathtt{nnz}(X) + \mathrm{poly}(d))\sum_{i=0}^{l-1}\log n_i \leq C(\mathtt{nnz}(X) + \mathrm{poly}(d))\log n \log\log n$$

$$\in O((\mathtt{nnz}(X) + \mathrm{poly}(d))\log n \log\log n).$$

Regarding the number of passes, note that for any $\eta > 0$, after $\log(\frac{1}{\eta})$ recursion steps, the leading term in the size of the coreset is as low as $n^{2^{-\log\frac{1}{\eta}}} = n^\eta$, after which we may arguably assume, that the coreset fits into memory. The algorithm thus takes $2\log(\frac{1}{\eta})$ streaming passes over the data before it turns to an internal memory algorithm. $\square$

# B  Material for the experimental section

Table 1: Absolute values of the negative log-likelihood $\mathcal{L}(\beta_{opt})$ at the optimal value $\beta_{opt}$ and mean running time $t_{opt}$ in seconds from the optimization task on the full data sets.

| DATA SET | $\mathcal{L}(\beta_{opt})$ | $t_{opt}$ |
|---|---|---|
| WEBB SPAM | 69,534.49 | 1,051.72 |
| COVERTYPE | 270,585.34 | 218.22 |
| KDD CUP '99 | 301,023.24 | 136.06 |

Figure 2: Each column shows the results for one data set comprising thirty different coreset sizes (depending on the individual size of the data sets). The plotted values are means and standard deviations taken over twenty independent repetitions of each experiment. The plots show the mean relative running times (upper row), the standard deviations of the relative log-likelihood errors (middle row) and standard deviations of the relative running times (lower row) of the three subsampling distributions, uniform sampling (blue), our QR derived distribution (red), and the $k$-means based distribution (green). All values are relative to the corresponding running times respectively optimal log-likelihood values of the optimization task on the full data set, see Table 1 (lower is better).

## C  Discussion of uniform sampling

As we have discussed in the lower bounds section 3, uniform sampling cannot help to build coresets of sublinear size for worst case instances. Actually this also holds for other techniques for solving logistic regression that rely on uniform subsampling, such as stochastic gradient descent (SGD).

We support this claim via a little experiment and some theoretical discussion on the following data set $X$ of size $m = 2n + 2$ in one dimension (plus intercept): The $-1$ class consists of one point at $-n$ and $n$ points at $1$, while class $+1$ consists of one point at $+n$ and $n$ points at $-1$. By symmetry of $\ell_1$-norms, it is straightforward to check that the data is $\mu$-complex for $\mu = 1$, and the optimal solution is $\hat{\beta} = 0$, which corresponds to $f(X\hat{\beta}) = \sum_{i=1}^{m} \ln(1 + \exp(0)) = m \ln(2) < m$. Our algorithms will thus find a coreset of sublinear size such that the optimal solution has a value of at most $(1 + \varepsilon)m$ with high probability.

A uniform sample of sublinear size misses the two points at $-n$ and $n$, since the probability to sample one of these is $\frac{1}{n+1}$. However, finding these points is crucial, since otherwise the remaining data is separable, which leads to a large $\beta$. Adding penalization is not a remedy. Figure 3 shows the results of running $1\,000$ independent repetitions of `sklearn.linear_model.SGDClassifier` in Python for logistic regression, with $\ell_2^2$-penalty enabled, on the data set with $n = 50\,000$. The boxplots show the resulting coefficients for the intercept $\beta_0$ and for the single dimension $\beta_1$. One might argue that the intercept term is close to $\hat{\beta}_0 = 0$, but for $\beta_1$, half of the values lie above the median of $107.06$ (red line) and still a quarter lies even above the upper quartile of $492.35$ (upper boundary of the box).

Note that assuming $\beta_0 = 0$ and $\beta_1 \gg (1 + \varepsilon)$, we have $f(X\beta) > 2n\beta_1 \gg (1 + \varepsilon)m$, since by construction

$$f(X\beta) = \sum_{i=1}^{m} \ln(1 + \exp(-y_i(\beta_0 + x_i\beta_1)))$$
$$\geq 2n \cdot \ln(1 + \exp(\beta_0 + \beta_1))$$
$$\geq 2n\beta_1.$$

This implies that the approximation ratio is $\frac{f(X\beta)}{f(X\hat{\beta})} \geq \frac{2n\beta_1}{m} = \frac{2n\beta_1}{2n+2} \overset{n \to \infty}{\longrightarrow} \beta_1$, which turned out very large in the experiment above, cf. Figure 3.

Figure 3: Boxplots of the solutions $\beta = (\beta_0, \beta_1)$ for logistic regression found by SGD in $1\,000$ independent runs on the considered data set $X$. The optimal solution is $\hat{\beta} = (0, 0)$. It can be seen that while the intercept term $\beta_0$ is reasonably close to $0$, the majority of runs result in considerably large values of $\beta_1$, which leads to a bad approximation ratio.