[Reviews · NeurIPS 2018]

Reviewer 1



The goal of this paper is to speed up logistic regression using a coreset based approach. The key idea is to “compress” the data set into a small fake set of points (called coreset) and to then train on that small set. The authors first show that, in general, no sublinear size coreset can exist. Then, they provide an algorithm that provides small summaries for certain data sets that satisfy a complexity assumption. Finally, they empirically compare that algorithm to two competing methods. Overall, the paper seems sound on the theoretical side; however, there are weaknesses in the related work and the experiments. If some of these concerns would be addressed in the rebuttal, I would be willing to upgrade my recommended score. Strengths: - The results seem to be correct. - In contrast to Huggins et al. (2016) and Tolochinsky & Feldman (2018), the coreset guarantee applies to the standard loss function of logistic regression and not to variations. - The (theoretical) algorithm (without the sketching algorithm for the QR decomposition) seems simple and practical. If space permits, the authors might consider explicitly specifying the algorithm in pseudo-code (so that practitioners do not have to extract it from the Theorems). - The authors include in the Supplementary Materials an example where uniform sampling fails even if the complexity parameter mu is bounded. - The authors show that the proposed approach obtains a better trade-off between error and absolute running time than uniform sampling and the approach by Huggins et al. (2016). Weaknesses: - Novelty: The sensitivity bound in this paper seems very similar to the one presented in [1] which is not cited in the manuscript. The paper [1] also uses a mix between sampling according to the data point weights and the l2-sampling with regards to the mean of the data to bound the sensitivity and then do importance sampling. Clearly, this paper treats a different problem (logistic regression vs k-means clustering) and has differences. However, this submission would be strengthened if the proposed approach would be compared to the one in [1]. In particular, I wonder if the idea of both additive and multiplicative errors in [1] could be applied in this paper (instead of restricting mu-complexity) to arrive at a coreset construction that does not require any assumptions on the data data set. [1] Scalable k-Means Clustering via Lightweight Coresets Olivier Bachem, Mario Lucic and Andreas Krause To Appear In International Conference on Knowledge Discovery and Data Mining (KDD), 2018. - Practical significance: The paper only contains a limited set of experiments, i.e., few data sets and no comparison to Tolochinsky & Feldman (2018). Furthermore, the paper does not compare against any non-coreset based approaches, e.g., SGD, SDCA, SAGA, and friends. It is not clear whether the proposed approach is useful in practice compared to these approaches. - Figure 1 would be much stronger if there were error bars and/or if there were more random trials that would (potentially) get rid of some of the (most likely) random fluctuations in the results.

Reviewer 2



This paper provides the first negative/positive results for constructing coresets in logistic regression. There are numerous results on coresets for regression problems but nothing was known for logistic regression. The authors first provide a negative result, meaning they show that it is not possible to construct a linear coreset for logistic regression in the general case. Then, under certain assumptions, they provide the first provably accurate coreset construction algorithm for linear regression. They support the strong theoretical results with experiments. Finally, i would recommend to the authors that they provide a detailed comparison of their results with this work https://arxiv.org/abs/1805.08571.

Reviewer 3



This paper provide new results on coresets (a reduced data set, which can be used as substitute for the large full data set), for use in logistic regression. Once the data is reduced, black-box logistic regression code can be run on the reduced set (now faster since the data is smaller) and the result is guaranteed to be close (with relative error i the loss function) to that of the full data set. Specifically, this paper (a) shows that no coreset of size o(n / log n) is possible, even if the coreset is not a strict subset of the original data. This is quite a strong negative result. (b) introduces a data parameter mu, which represents how balanced and separable the dataset can be (the smaller the better). If the input data has this data large, this often corresponds to ill-posed problems (data is not well-separable, or heavily imbalanced), and hence logistic regression is not a good fit. (c) provides an algorithm, using a variant of sensitivity sampling, that if mu is small, is guaranteed to provide a small coreset with strong approximation guarantees. The algorithm essential reduces to a (approximate) QR decomposition to learn weights, and the sampling. It empirically performs favorably to uniform sample, and a previous approach which essentially performs approximate k-means++ clustering. This is a clearly written and precise result on a topic of significant interest. The paper provides a careful and informed view of prior work in this area, and builds on it in careful ways. While the use of mu, may not be the final result on this topic, this is a real and important concrete benchmark. Suggestions and minor concerns: (these are secondary to the main contributions) * the dependence in the coreset size on mu, d, and eps are large when the size is logarithmic in the size n: mu^6, 1/eps^4, d^4. plus some log factors. (alternatively, the size requires sqrt{n} space with smaller values in other parameters : mu, 1/eps^2, d^{3/2}) * The paper says mu can be estimated efficiently (in poly(nd) time). In this context, this does not seem very efficient. How should this be estimated in the context of the near-linear runtime algorithms? What are the values for the data sets used? * it would be interesting to see how this compared to sensitivity sampling-based approaches (e.g., Barger and Feldman) designed for clustering applications.